# DIVA: Dataset Derivative of a Learning Task

**Yonatan Dukler**[1,2]*, **Alessandro Achille**[1], **Giovanni Paolini**[1], **Avinash Ravichandran**[1],
**Marzia Polito**[1], **Stefano Soatto**[1]

[1] Amazon Web Services,
`{aachille, paoling, ravinash, mpolito, soattos}@amazon.com`
[2] Department of Mathematics,
University of California, Los Angeles
`ydukler@math.ucla.edu`

## Abstract

We present a method to compute the derivative of a learning task with respect to a dataset. A learning task is a function from a training set to the validation error, which can be represented by a trained deep neural network (DNN). The "dataset derivative" is a linear operator, computed around the trained model, that informs how perturbations of the weight of each training sample affect the validation error, usually computed on a separate validation dataset. Our method, DIVA (Differentiable Validation) hinges on a closed-form differentiable expression of the leave-one-out cross-validation error around a pre-trained DNN. Such expression constitutes the dataset derivative. DIVA could be used for dataset auto-curation, for example removing samples with faulty annotations, augmenting a dataset with additional relevant samples, or rebalancing. More generally, DIVA can be used to optimize the dataset, along with the parameters of the model, as part of the training process without the need for a separate validation dataset, unlike bi-level optimization methods customary in AutoML. To illustrate the flexibility of DIVA, we report experiments on sample auto-curation tasks such as outlier rejection, dataset extension, and automatic aggregation of multi-modal data.

## 1 Introduction

Consider the following seemingly disparate questions. *(i) Dataset Extension:* Given a relatively small training set, but access to a large pool of additional data, how to select from the latter samples to augment the former? *(ii) Dataset Curation:* Given a potentially large dataset riddled with annotation errors, how to automatically reject such outlier samples? *(iii) Dataset Reweighting:* Given a finite training set, how to reweight the training samples to yield better generalization performance?

These three are examples of *Dataset Optimization*. In order to solve this problem with differentiable programming, one can optimize a loss of the model end-to-end, which requires *differentiating the model's loss with respect to the dataset.* Our main contribution is an efficient method to compute such a *dataset derivative*. This allows learning an importance weight $\alpha_i$ for each datum in a training dataset $\mathcal{D}$, extending the optimization from the weights $\mathbf{w}$ of a parametric model such as a deep neural network (DNN), to also include the weights of the dataset.

As illustrated in the following diagram, standard optimization in machine learning works by finding the weights $\mathbf{w}_\alpha$ that minimize the training loss $L_{\text{train}}(\mathbf{w}, D_\alpha) = \sum_i \alpha_i \ell(f_{\mathbf{w}}(\mathbf{x}_i), y_i)$ on a given (weighted) dataset $D_\alpha$ (dark box). We solve a more general learning problem (light box) by jointly optimizing the dataset $\mathcal{D}_\alpha$ in addition to $\mathbf{w}$. To avoid the trivial solution $\alpha = 0$, it is customary in AutoML to optimize $\mathcal{D}_\alpha$ by minimizing the validation error computed on a disjoint dataset. This makes for inefficient use of the data, which has to be split between training and validation sets. Instead, we leverage a closed-form expression of the leave-one-out cross-validation error to jointly optimize the model and data weights during training, without the need to create a separate validation set.

---

*Work conducted at Amazon Web Services.

$$D_\alpha \longrightarrow L_{\text{train}}(w, D_\alpha) \longrightarrow w_\alpha \longrightarrow L_{\text{val}}(w_\alpha)$$

The intermediate block in the diagram (which finds the optimal weights $\mathbf{w}_\alpha$ for the the training loss on $\mathcal{D}_\alpha$) is usually non-differentiable with respect to the dataset, or the derivative is prohibitively expensive to compute. DIVA leverages recent progress in deep learning linearization Achille et al. (2020), to derive a closed-form expression for the derivative of the final loss (validation error) with respect to the dataset weights. In particular, Achille et al. (2020) have shown that, by replacing cross-entropy with least-squares, replacing ReLu with leaky-ReLu, and performing suitable pre-conditioning, the linearized model performs on par with full non-linear fine-tuning. We also leverage a classical result to compute the leave-one-out loss of a linear model in closed-form (Rifkin & Lippert, 2007; Green & Silverman, 1993). This allows us to optimize the LOO loss without requiring a separate validation set, setting DIVA apart from bi-level optimization customary in AutoML.

To illustrate the many possible uses of the dataset derivative, we run experiments with a simplified version of DIVA to cleanup a dataset of noisy annotations, to extend a training set with additional data from an external pool, identify meaningful data augmentation, and to perform multi-modal expansion using a CLIP model (Radford et al., 2021).

Rather than using the full linearization of the model derived by Achille et al. (2020), we restrict the gradient to its last layer, cognizant that we are not exploiting the full power of LQF and thereby obtaining only a lower-bound of performance improvement. Despite that restriction, our results show consistent improvements from dataset optimization, at the modest computational cost of a forward pass over the dataset to optimize the importance weights.

To summarize, our main contributions are:
1. We introduce a method to compute the dataset derivative in closed form, DIVA.
2. We illustrate the use of DIVA to perform dataset optimization by minimizing directly the leave-one-out error without the need for an explicit validation dataset.
3. We perform experiments with a simplified model that, despite not using the full power of the linearization, shows consistent improvements in dataset extension, re-weighting, outlier rejection and automatic aggregation of multi-modal data.

Our method presents several limitations. The dataset derivative of a learning task is computed around a point represented by a pre-trained model. It only allows local optimization around this point. Moreover, we only compute a restriction of the linearization to the dimensions spanned by the last few layers. In general, this yields suboptimal results compared to full global optimization from scratch, if one could compute that at scale. Nonetheless, the linearized setting is consistent with the practice of fine-tuning pre-trained models in light of the results of Achille et al. (2020), see also (Radford et al., 2021; Rezende et al., 2017; Mormont et al., 2018; Hosny et al., 2018).

## 2 RELATED WORK

**AutoML.** State of the art performance in image classification tasks often relies on large amount of human expertise in selecting models and adjusting the training settings for the task at hand (Li et al., 2020a). Automatic machine learning (AutoML) (Feurer et al., 2019; He et al., 2021) aims to automate model selection (Cawley & Talbot, 2010) and the training settings by instead using meta-algorithms for the different aspects of the learning settings. Such methods follow a bi-level optimization framework, optimizing the training settings in the outer level, and traditional model optimization in the inner level (Jenni & Favaro, 2018). AutoML has focused on achieving better results via automatic model selection (Deshpande et al., 2021; Feurer et al., 2019) including neural architecture search (NAS) (Zoph & Le, 2016; Elsken et al., 2019; Liu et al., 2019). Other important AutoML topics include hyper-parameter selection (Li et al., 2017; Akiba et al., 2019) and data augmentation (Cubuk et al., 2018; Lim et al., 2019; Chen et al., 2020; Behl et al., 2020), which are closer to our settings of optimizing the dataset weights. Since the main signal for a model's performance is the final validation loss, which requires full optimization of the model for each evaluation, AutoML approaches often incur a steep computational costs. Alternatively, other methods

follow alternating optimization of the criteria, such as the work of Ren et al. (2018) that approximates full network optimization with a single SGD step to learn to reweight the training set dynamically. *Differentiable AutoML* alleviates outer-optimization costs while optimizing the final validation error via differentiable programming, by utilizing proxy losses and continuous relaxation that enable differentiation. Different approaches to differentiable AutoML include differentiable NAS (Liu et al., 2018; Wu et al., 2019), data augmentation (Liu et al., 2021; Li et al., 2020b), and hyper-parameter optimization (Andrychowicz et al., 2016). The DIVA dataset derivative follows the differentiable AutoML framework by enabling direct optimization of the dataset with respect to the final validation error of the model.

**Importance sampling.** While our dataset optimization problem may seem superficially similar to importance sampling, the optimization objective is different. Importance sampling aims to reweight the training set to make it more similar to the test distribution or to speed up convergence. On the other hand, DIVA objective is to optimizes a validation loss of the model, even if this requires making the training distribution significantly different from the testing distribution. Importance sampling methods have a long history in the MCMC machine learning literature where the sampling is conditioned on the predicted importance of samples (Metropolis & Ulam, 1949; Liu, 2008). In deep learning, importance sampling methods have been studied theoretically for linearly-separable data (Byrd & Lipton, 2019) and recently in more generality (Xu et al., 2021). Furthermore, there exist many importance sampling heuristics in deep learning training including different forms of hard sample mining (Shrivastava et al., 2016; Xue et al., 2019; Chang et al., 2017), weighting based on a focal loss (Lin et al., 2017), re-weighting for imbalance, (Cui et al., 2019; Huang et al., 2019; Dong et al., 2017) and gradient based scoring (Li et al., 2019). We emphasize that DIVA's optimization of the sample weights is not based on a heuristic but is rather a differentiable AutoML method driven by optimization of a proxy of the test error. Further, DIVA allows optimization of the dataset weights with respect to an arbitrary loss and also allows for dataset extension computation.

**LOO based optimization.** Leave-one-out cross validation is well established in statistical learning (Stone, 1977). In ridge regression, the LOO model predictions for the validation samples have a closed-form expression that avoids explicit cross validation computation (Green & Silverman, 1993; Rifkin & Lippert, 2007) enabling efficient and scalable unbiased estimate of the test error. Efficient LOO has been widely used as a criterion for regularization (Pedregosa et al., 2011; Quan et al., 2010; Birattari et al., 1999; Thapa et al., 2020), hyper-parameter selection (Hwang & Shim, 2017) and optimization (Wen et al., 2008). Most similar to our dataset derivative are methods that: (1) optimize a restricted set of parameters, such as kernel bandwidth, in weighted least squares (Cawley, 2006; Hong et al., 2007) (2) locally weighted regression methods (*memorizing regression*) (Atkeson et al., 1997; Moore et al., 1992), or (3) methods that measure the impact of samples based on LOO predictions (Brodley & Friedl, 1999; Nikolova et al., 2021).

**Dataset selection & sample impact measures.** Koh & Liang (2017) measure the effect of changes of a training sample weight on a final validation loss through per-sample weight gradients, albeit without optimizing the dataset and requiring a separate validation set. Their proposed expression for the per-sample gradient, however, does not scale easily to our problem of dataset optimization. In contrast, in proposition 3 we introduce an efficient closed-form expression for the derivative of the whole datasets. Moreover, in proposition 3, we show how to optimize the weights with respect to a cross-validation loss which does not require a separate set. In Pruthi et al. (2020), the authors present a sample-impact measure for interpretability based on a validation set; for dataset extension, Yan et al. (2020) presents a coarse dataset extension method based on self-supervised learning. Dataset distillation and core set selection methods aim to decrease the size of the dataset (Wang et al., 2018) by selecting a representative dataset subset (Hwang et al., 2020; Jeong et al., 2020; Coleman et al., 2019; Joneidi et al., 2020; Trichet & Bremond, 2018; Killamsetty et al., 2021). While DIVA is capable of removing outliers, in this work we do not approach dataset selection from the perspective of making the dataset more computationally tractable by reducing the number of samples.

## 3 METHOD

In supervised learning, we use a parametrized model $f_{\mathbf{w}}(\mathbf{x})$ to predict a target output $y$ given an input $\mathbf{x}$ coming from a joint distribution $(\mathbf{x}, y) \sim \mathcal{T}$. Usually, we are given a training set $\mathcal{D} = \{(\mathbf{x}_i, y_i)\}_{i=1}^{N}$ with samples $(x, y)$ assumed to be independent and identically distributed (i.i.d.) according to $\mathcal{T}$.

The training set $\mathcal{D}$ is then used to assemble the empirical risk for some per-sample loss $\ell$,

$$L_{\text{train}}(\mathbf{w}; \mathcal{D}) = \sum_{i=1}^{N} \ell(f_{\mathbf{w}}(\mathbf{x}_i), y_i),$$

which is minimized to find the optimal model parameters $\mathbf{w}_{\mathcal{D}}$:

$$\mathbf{w}_{\mathcal{D}} = \underset{\mathbf{w}}{\operatorname{argmin}}\, L_{\text{train}}(\mathbf{w}; \mathcal{D}).$$

The end goal of empirical risk minimization is that weights will also minimize the *test loss*, computed using a separate test set. Nonetheless $\mathcal{D}$ is often biased and differs from the distribution $\mathcal{T}$. In addition, from the perspective of optimization, different weighting of the training loss samples can enable or inhibit good learning outcomes of the task $\mathcal{T}$ (Lin et al., 2017).

**Dataset Optimization.** In particular, it may not be the case that sampling the training set $\mathcal{D}$ i.i.d. from $\mathcal{T}$ is the best option to guarantee generalization, nor it is realistic to assume that $\mathcal{D}$ is a fair sample. Including in-distribution samples that are too difficult may negatively impact the optimization, while including certain out-of-distribution examples may aid the generalization on $\mathcal{T}$. It is not uncommon, for example, to improve generalization by training on a larger dataset containing out-of-distribution samples coming from other sources, or generating out-of-distribution samples with data augmentation. We call Dataset Optimization the problem of finding the optimal subset of samples, real or synthetic, to include or exclude from a training set $\mathcal{D}$ in order to guarantee that the weights $\mathbf{w}_{\mathcal{D}}$ trained on $\mathcal{D}$ will generalize as much as possible.

**Differentiable Dataset Optimization.** Unfortunately, a naïve brute-force search over the $2^N$ possible subsets of $\mathcal{D}$ is unfeasible. The starting idea of DIVA is to instead solve a more general continuous optimization problem that can however be optimized end-to-end. Specifically, we parameterize the choice of samples in the augmented dataset through a set of non-negative continuous sample weights $\alpha_i$ which can be optimized by gradient descent along with the weights of the model. Let $\alpha = (\alpha_1, \ldots, \alpha_N)$ be the vector of the sample weights and denote the corresponding weighted dataset by $\mathcal{D}_\alpha$. The training loss on $\mathcal{D}_\alpha$ is then defined as:

$$L_{\mathcal{D}}(\mathbf{w}; \mathcal{D}_\alpha) = \sum_{i=1}^{N} \alpha_i\, \ell(f_{\mathbf{w}}(\mathbf{x}_i), y_i). \tag{1}$$

Note that if all $\alpha_i$'s are either 0 or 1, we are effectively selecting only a subset of $\mathcal{D}$ for training. As we will show, this continuous generalization allows us to optimize the sample selection in a differentiable way. In principle, we would like to find the sample weights $\alpha^* = \operatorname{argmin}_\alpha L_{\mathcal{D}_{\text{test}}}(\mathbf{w}_\alpha)$ that lead to the best generalization. Since we do not have access to the test data, in practice this translates to optimizing $\alpha$ with respect to an (unweighted) validation loss $L_{\text{val}}$:

$$\alpha^* = \operatorname{argmin}_\alpha L_{\text{val}}(\mathbf{w}_\alpha).$$

We can, of course, compute a validation loss using a separate validation set. However, as we will see in Section 3.3, we can also use a leave-one-out cross-validation loss directly on the training set, without any requirement of a separate validation set.

In order to efficiently optimize $\alpha$ by gradient-descent, we need to compute the dataset derivative $\nabla_\alpha L_{\text{val}}(\mathbf{w}_\alpha)$. By the chain rule, this can be done by computing $\nabla_\alpha \mathbf{w}_\alpha$. However, the training function $\alpha \to \mathbf{w}_\alpha$ that finds the optimal weights $\mathbf{w}_\alpha$ of the model given the sample weights $\alpha$ may be non-trivial to differentiate or may not be differentiable at all (for example, it may consist of thousands of steps of SGD). This would prevent us from minimizing $\alpha$ end-to-end.

In the next section, we show that if, instead of linearizing the $\mathbf{w}_\alpha$ end-to-end in order to compute the derivative, we linearize the model *before* the optimization step, the derivative can both be written in closed-form and computed efficiently, thus giving us a tractable way to optimize $\alpha$.

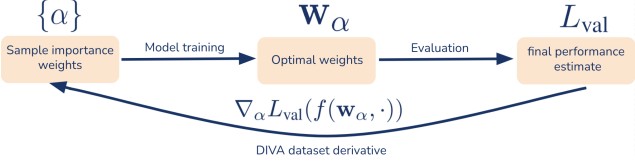

Figure 1: The DIVA dataset derivative is computed end-to-end from the final validation loss

### 3.1 LINEARIZATION

In real-world applications, the parametric model $f_{\mathbf{w}}(\mathbf{x})$ is usually a deep neural network. Recent work (Achille et al., 2020; Mu et al., 2020) have shown that in many cases, a deep neural network can be transformed to an equivalent linear model that can be trained on a simple quadratic loss and still reach a performance similar to the original model. Given a model $f_{\mathbf{w}}(\mathbf{x})$, let $\mathbf{w}_0$ denote an initial set of weights, For example, $\mathbf{w}_0$ could be obtained by pre-training on a large dataset such as ImageNet (if the task is image classification). Following Achille et al. (2020); Mu et al. (2020), we consider a linearization $f_{\mathbf{w}}^{\text{lin}}(\mathbf{x})$ of the network $f_{\mathbf{w}}(\mathbf{x})$ given by the first-order Taylor expansion of $f_{\mathbf{w}}(\mathbf{x})$ around $\mathbf{w}_0$:

$$f_{\mathbf{w}}^{\text{lin}}(\mathbf{x}) = f_{\mathbf{w}_0}(\mathbf{x}) + \nabla_{\mathbf{w}} f_{\mathbf{w}_0}(\mathbf{x}) \cdot (\mathbf{w} - \mathbf{w}_0). \tag{2}$$

Intuitively, if fine-tuning does not move the weights much from the initial pre-trained weights $\mathbf{w}_0$, then $f_{\mathbf{w}}^{\text{lin}}(\mathbf{x})$ will remain a good approximation of the network while becoming linear in $\mathbf{w}$ (but still remaining highly non-linear with respect to the input $\mathbf{x}$). Effectively, this is equivalent to training a linear classifier using the gradients $\mathbf{z}_i := \nabla_{\mathbf{w}} f_{\mathbf{w}_0}(\mathbf{x}_i)$ as features (Mu et al., 2020).

Although $f_{\mathbf{w}}^{\text{lin}}(\mathbf{x})$ is a linear model, the optimal weights $\mathbf{w}_\alpha$ may still be a complex function of the training data, depending on the loss function used. Achille et al. (2020) showed that equivalent performance can be obtained by replacing the empirical cross-entropy with the regularized least-squares loss:

$$L_{\mathcal{D}}(\mathbf{w}) = \sum_{i=1}^{N} \|f_{\mathbf{w}}^{\text{lin}}(\mathbf{x}) - \mathbf{y}_i\|^2 + \lambda \|\mathbf{w}\|^2 \tag{3}$$

where $\mathbf{y}$ denotes the one-hot encoding vector of the label $y_i$. In Achille et al. (2020), it is shown that linearized models are equivalent from the standpoint of performance on most standard tasks and classification benchmarks, and better in the low-data regime, which is where the problem of "dataset augmentation" is most relevant. The advantage of using this loss is that the optimal weights $\mathbf{w}^*$ can now be written in closed-form as

$$\mathbf{w}^* = (\mathbf{Z}^\top \mathbf{Z} + \lambda \mathbf{I})^{-1} \mathbf{Z}^\top (\mathbf{Y} - f_{\mathbf{w}_0}(\mathbf{X})), \tag{4}$$

where $\mathbf{Z} = [\mathbf{z}_1, \ldots, \mathbf{z}_N]$ is the matrix of the Jacobians $\mathbf{z}_i = \nabla_{\mathbf{w}} f_{\mathbf{w}_0}(\mathbf{x}_i)$. While our method can be applied with no changes to linearization of the full network, for simplicity in our experiments we restrict to linearizing only the last layer of the network. This is equivalent to using the network as a fixed feature extractor and training a linear classifier on top the last-layer features, that is, $\mathbf{z}_i = f_{\mathbf{w}_0}^{L-1}(\mathbf{x}_i)$ are the features at the penultimate layer.

### 3.2 COMPUTATION OF THE DATASET DERIVATIVE

We now show that for linearized models we can compute the derivative $\nabla_\alpha \mathbf{w}_\alpha$ in closed-form. For the $\alpha$-weighted dataset, the objective in eq. (3) with $L_2$ loss for the linearized model is written as,

$$\mathbf{w}_\alpha = \operatorname*{argmin}_{\mathbf{w}} L_{\mathcal{D}}(\mathbf{w}; \mathcal{D}_\alpha) = \operatorname*{argmin}_{\mathbf{w}} \sum_{i=1}^{N} \alpha \|\mathbf{w}^\top \mathbf{z}_i - \mathbf{y}_i\|^2 + \lambda \|\mathbf{w}\|^2. \tag{5}$$

where $\mathbf{z}_i = \nabla_{\mathbf{w}} f_{\mathbf{w}_0}(\mathbf{x}_i)$ as in the previous section. Note that $\alpha \|\mathbf{w}^\top \mathbf{z}_i - \mathbf{y}_i\|^2 = \|\mathbf{w}^\top \mathbf{z}_i^\alpha - \mathbf{y}_i^\alpha\|$, where $\mathbf{z}_i^\alpha := \sqrt{\alpha} \mathbf{z}_i$ and $\mathbf{y}_i^\alpha := \sqrt{\alpha} \mathbf{y}_i$. Using this, we can reuse eq. (4) to obtain the following closed-form solution for $\mathbf{w}_\alpha$:

$$\mathbf{w}_\alpha = (\mathbf{Z}^\top \mathbf{D}_\alpha \mathbf{Z} + \lambda \mathbf{I})^{-1} \mathbf{Z}^\top \mathbf{D}_\alpha \mathbf{Y}, \tag{6}$$

where we have taken $\mathbf{D}_\alpha = \operatorname{diag}(\alpha)$. In particular, note that $\mathbf{w}_\alpha$ is now a differentiable function of $\alpha$. The following proposition gives a closed-form expression for the derivative.

**Proposition 1** (Model-Dataset Derivative $\nabla_\alpha \mathbf{w}_\alpha$). *For the ridge regression problem equation 5 and $\mathbf{w}_\alpha$ defined as in equation 6, define*

$$\mathbf{C}_\alpha = (\mathbf{Z}^\top \mathbf{D}_\alpha \mathbf{Z} + \lambda \mathbf{I})^{-1}. \tag{7}$$

*Then the Jacobian of $\mathbf{w}_\alpha$ with respect to $\alpha$ is given by*

$$\nabla_\alpha \mathbf{w}_\alpha = \mathbf{Z} \mathbf{C}_\alpha \circ \left( (\mathbf{I} - \mathbf{Z} \mathbf{C}_\alpha \mathbf{Z}^\top \mathbf{D}_\alpha) \mathbf{Y} \right), \tag{8}$$

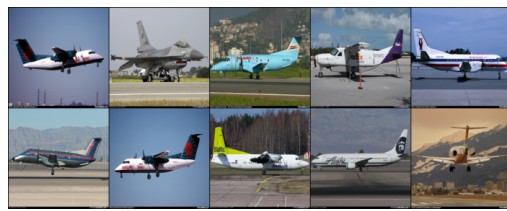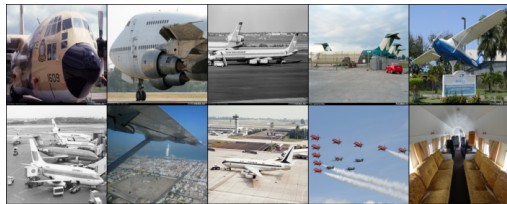

Figure 2: **Examples of the reweighting done by DIVA. (Left)** Samples from the FGVC Aircraft classification dataset that are up-weighted by DIVA and **(Right)** samples that are down-weighted because they increase the test error. Down-weighted samples tend to have planes in non-canonical poses, multiple planes, or not enough information to classify the plane correctly.

Where we write $\mathbf{A} \circ \mathbf{B} \in \mathbb{R}^{n \times m \times k}$ for the batch-wise outer product of $\mathbf{A} \in \mathbb{R}^{n \times m}$ and $\mathbf{B} \in \mathbb{R}^{n \times k}$ along the common dimension $k$, i.e., $(\mathbf{A} \circ \mathbf{B})_{ijk} = a_{ij}b_{ik}$

The Jacobian $\nabla_\alpha \mathbf{w}_\alpha$ would be rather large to compute explicitly. Fortunately, the end-to-end gradient $L_{\text{val}}(\mathbf{w}_\alpha)$ if the final validation loss can still be computed efficiently, as we now show. Given a validation dataset $\mathcal{D}_{\text{val}}$, the validation loss is:

$$L_{\text{val}}(\mathbf{w}_\alpha) = \sum_{(\mathbf{x}_i, y_i) \in \mathcal{D}_{\text{val}}} \ell(f_{\mathbf{w}_\alpha}(\mathbf{x}_i), y_i). \tag{9}$$

The following gives the expression to optimize $\alpha$ end-to-end with respect to the validation loss.

**Proposition 2** (Validation Loss Dataset Derivative). *Define $\mathbf{L}$ as the loss function derivative with respect to the network outputs as,*

$$\mathbf{L} = \left[ \frac{\partial \ell}{\partial f}(f(\mathbf{x}_1), y_1), \cdots \frac{\partial \ell}{\partial f}(f(\mathbf{x}_N), y_N) \right]$$

*Then the dataset derivative importance weights with respect to final validation is given by*

$$\nabla_\alpha L_{val}(\mathbf{w}_\alpha) = \mathbf{Z}\mathbf{C}_\alpha \mathbf{Z}^\top \times \left( \mathbf{L}^\top \mathbf{Y}^\top (\mathbf{I} - \mathbf{D}_\alpha \mathbf{Z} \mathbf{C}_\alpha \mathbf{Z}^\top) \right). \tag{10}$$

### 3.3 LEAVE-ONE-OUT OPTIMIZATION

$\{\alpha\}$ It is common in AutoML to optimize the hyper-parameters with respect to a separate validation set. However, using a separate validation may not be practical in limited data settings, which are a main focus of dataset optimization. To remedy this, we now show that we can instead optimize $\alpha$ by minimizing a leave-one-out cross-validation loss that only requires a training set:

$$L_{\text{LOO}}(\alpha) = \sum_{i=1}^N \ell(f_{\mathbf{w}_\alpha^{-i}}(\mathbf{x}_i), y_i), \tag{11}$$

where $\mathbf{w}_\alpha^{-i}$ are the optimal weights obtained by training with the loss eq. (5) on the entire dataset $\mathcal{D}$ *except* for the $i$-th sample $(\mathbf{x}_i, y_i)$. This may seem counter-intuitive, since we are optimizing the weights of the training samples using a validation loss defined on the training set itself. It is useful to recall that $\mathbf{w}_\alpha^{-i}$ minimizes the $\alpha$-weighted $L_2$ loss on the training set (minus the $i$-th example):

$$\mathbf{w}_\alpha^{-i} = \arg\min_{\mathbf{w}} L_{\mathcal{D}}^{-i}(w, \mathcal{D}_\alpha) = \arg\min_{\mathbf{w}} \sum_{j \neq i} \alpha_j \|f_w(x_j) - y_j\|^2 + \lambda \|\mathbf{w}\|^2. \tag{12}$$

Meanwhile, $\alpha$ minimizes the *unweighted* validation loss in eq. (11). This prevents the existence of degenerate solutions for $\alpha$.

Computing $L_{\text{LOO}}$ naively would require training $n$ classifiers, but fortunately, in the case of a linear classifier with the $L_2$ loss, a more efficient closed-form solution exists (Green & Silverman, 1993; Rifkin & Lippert, 2007). Generalizing those results to the case of a weighted loss, we are able to derive the following expression.

**Proposition 3.** *Define*

$$\mathbf{R}_\alpha = \mathbf{Z}^\top \sqrt{\mathbf{D}_\alpha} (\mathbf{Z}^\top \mathbf{D}_\alpha \mathbf{Z} + \lambda \mathbf{I})^{-1} \sqrt{\mathbf{D}_\alpha} \mathbf{Z}$$

*Then $\alpha$-weighted LOOV predictions defined in eq. (12) admit a closed-form solution:*

$$f_{\mathbf{w}_\alpha^{-i}}(\mathbf{z}_i) = \left[ \frac{\mathbf{R}_\alpha\sqrt{\mathbf{D}_\alpha}\mathbf{Y} - diag(\mathbf{R}_\alpha)\sqrt{\mathbf{D}_\alpha}\mathbf{Y}}{diag(\sqrt{\mathbf{D}_\alpha} - \sqrt{\mathbf{D}_\alpha}\mathbf{R}_\alpha)} \right]_i, \tag{13}$$

*where $diag(\mathbf{A}) = [a_{11}, \ldots, a_{nn}]$ denotes the vector containing the diagonal of $\mathbf{A}$, and the division between vectors is element-wise.*

Note that the prediction $f_{\mathbf{w}_\alpha^{-i}}(\mathbf{z}_i)$ on the $i$-th sample when training on all the other samples is a differentiable function of $\alpha$. Composing eq. (13) in eq. (11), we compute the derivative $\nabla_\alpha L_{\text{LOO}}(\alpha)$, which allows us to optimize the cross-validation loss with respect to the sample weights, without the need of a separate validation set. We give the closed-form expression for $\nabla_\alpha L_{\text{LOO}}(\alpha)$ in the Appendix.

### 3.4 DATASET OPTIMIZATION WITH DIVA

We can now apply the closed-form expressions for $\nabla_\alpha L_{\text{val}}(\alpha)$ and $\nabla_\alpha L_{\text{LOO}}(\alpha)$ for differentiable dataset optimization. We describe the optimization using $L_{\text{val}}$, but the same applies to $L_{\text{LOO}}$.

**DIVA Reweight.** The basic task consists in reweighting the samples of an existing dataset in order to improve generalization. This can curate a dataset by reducing the influence of outliers or wrong labels, or by reducing possible imbalances. To optimize the dataset weights, we use gradient descent in the form:

$$\alpha \leftarrow \alpha - \eta \nabla_\alpha L_{\text{val}}. \tag{14}$$

It is important to notice that $L_{\text{val}}$ is an unbiased estimator of the test loss only at the first step, hence optimizing using eq. (14) for multiple steps can lead to over-fitting (see Appendix). Therefore, we apply only 1-3 gradient optimization steps with a relatively large learning rate $\eta \simeq 0.1$. This early stopping both regularizes the solution and decreases the wall-clock time required by the method. We initialize $\alpha$ so that $\alpha_i = 1$ for all samples.

**DIVA Extend.** The dataset gradient also allows us to extend an existing dataset. Given a core dataset $\mathcal{D} = \{(\mathbf{x}_i, y_i)\}_{i=1}^N$ and an external (potentially noisy) data pool $\mathcal{E} = \{(\hat{\mathbf{x}}_i, \hat{y}_i)\}_{i=N+1}^{N+M}$, we want to find the best samples from $\mathcal{E}$ to add to $\mathcal{D}$. For this we merge $\mathcal{D}$ and $\mathcal{E}$ in a single dataset and initialize $\alpha$ such that $\alpha_i = 1$ for samples of $\mathcal{D}$ and $\alpha_i = 0$ for samples of $\mathcal{E}$ (so that initially the weighted dataset matches $\mathcal{D}$). We then compute $\nabla_\alpha L_{\text{val}}(\alpha)$ to find the top $k$ samples of $\mathcal{E}$ that have the largest negative value of $\nabla_\alpha L_{\text{val}}(\alpha)_i$, i.e., the samples that would give the largest reduction in validation error if added to the training set and add them to $\mathcal{D}$. This is repeated until the remaining samples in $\mathcal{E}$ all have positive value for the derivative (adding them would not further improve the performance).

**Detrimental sample detection.** The $i$-th component of $\nabla_\alpha L_{\text{val}}$ specifies the influence of the $i$-th sample on the validation loss. In particular, $(\nabla_\alpha L_{\text{val}})_i > 0$ implies that the sample increases the validation loss, hence it is detrimental (e.g., it is mislabeled or overly represented in the dataset). We can select the set of detrimental examples by thresholding $\nabla_\alpha L_{\text{val}}$:

$$\text{Detrimental}(\epsilon) = \{i : (\nabla_\alpha L_{\text{val}})_i \geq \epsilon\}. \tag{15}$$

## 4 RESULTS

For our models we use standard residual architectures (ResNet) models pre-trained on ImageNet (Deng et al., 2009) and Places365 (Zhou et al., 2017). For our experiments on dataset optimization we consider datasets that are smaller than the large scale datasets used for pre-training as we believe they reflect more realistic conditions for dataset optimization. For our experiments we use the CUB-200 (Welinder et al., 2010), FGVC-Aircraft, (Maji et al., 2013), Stanford Cars (Krause et al., 2013), Caltech-256 (Griffin et al., 2007), Oxford Flowers 102 (Nilsback & Zisserman, 2008), MIT-67 Indoor (Quattoni & Torralba, 2009), Street View House Number (Netzer et al., 2011), and the Oxford Pets (Parkhi et al., 2012) visual recognition and classification datasets. In all experiments, we use the network as a fixed feature extractor, and train a linear classifier on top of the network features using the weighted $L_2$ loss eq. (5) and optimize the $\alpha$ weights using DIVA.

**Dataset AutoCuration.** We use DIVA Reweight to optimize the importance weights of samples from several fine-grain classification datasets. While the datasets have already been manually curated

| Dataset | Original | DIVA Reweight | Chang et al. (2017)[†] | Ren et al. (2018)[*] | Gain |
|---|---|---|---|---|---|
| Aircrafts | 57.58 | **54.64** | 70.48 | 81.82 (80.62) | +2.94 |
| Cub-200 | 39.30 | **36.93** | 57.85 | 72.55 (75.35) | +2.36 |
| MIT Indoor-67 | 32.54 | **31.27** | 37.84 | 64.48 (58.06) | +1.27 |
| Oxford Flowers | 20.23 | **19.16** | 22.82 | 48.80 (55.46) | +1.07 |
| Stanford Cars | 58.91 | **56.31** | 75.87 | 83.09 (84.50) | +2.56 |
| Caltech-256 | 23.98 | **21.29** | 37.52 | 58.44 (52.77) | +2.69 |

Table 1: Test error of DIVA Reweight to curate several fine-grain classification datasets. We use a ResNet-34 pretrained on ImageNet as feature extractor and train a linear classifier on top of the last layer. Note that DIVA Reweight can improve performance even on curated and noiseless datasets whereas other reweighting methods based on hard-coded rules may be detrimental in this case.

by experts to exclude out-of-distribution or mislabeled examples, we still observe that in all cases DIVA can further improve the test error of the model (Table 1). To understand how DIVA achieves this, in Figure 2 we show the most up-weighted (left) and down-weighted (right) examples on the FGVC Aircraft classification task Maji et al. (2013). We observe that DIVA tends to give more weight to clear, canonical examples, while it detects as detrimental (and hence down-weights) examples that contain multiple planes (making the label uncertain), or that do not clearly show the plane, or show non-canonical poses. We compare DIVA Reweight with two other re-weighting approaches: Ren et al. (2018), that applies re-weighting using information extracted from a separate validation gradient step, and Chang et al. (2017), which reweighs based on the uncertainty of each prediction (threshold-closeness weighting scheme). For Ren et al. (2018), we set aside 20% of the training samples as validation for the reweight step, but use all samples for the final training (in parentheses). We notice that both baselines under-perform with respect to DIVA on noiseless datasets.

**Dataset extension.** We test the capabilities of DIVA Extend to extend a dataset with additional samples of the distribution. In Figure 4 and Table 2 (in the Appendix), we observe that DIVA is able to select the most useful examples and reaches an optimal performance generalization error using significantly less samples than the baseline uniform selection. Moreover, we notice that DIVA identifies a smaller subset of samples that provides better test accuracy than adding all the pool samples to the training set.

**Detrimental sample detection.** To test the ability of DIVA to detect training samples that are detrimental for generalization, we introduce wrong labels in the dataset. In Section 3.4 we suggest detecting detrimental examples by finding where is $\nabla_\alpha L_{\text{LOO}}(\alpha)_i$ positive. To verify this, in Figure 3 we plot the histogram of the derivatives for correct and mislabeled examples. We observe that most mislabeled examples have positive derivative. In particular, we can directly classify an example as mislabeled if the derivative is positive. In Figure 3 we report the F1 score and AUC obtained in a mislabeled sample detection task using the DIVA gradients.

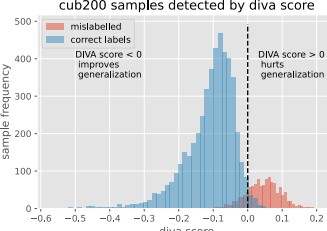

| Dataset | F1-score ($\epsilon = 0$) | AUC |
|---|---|---|
| Cub200 | 0.87 | 0.98 |
| Aircrafts | 0.68 | 0.90 |
| MIT Indoor-67 | 0.86 | 0.98 |
| Stanford Cars | 0.75 | 0.93 |
| Caltech-256 | 0.92 | 0.99 |
| Oxford Flowers | 0.83 | 0.97 |

Figure 3: **(Left)** Distribution of LOO DIVA gradients for correctly labelled and mislabelled samples in CUB-200 dataset (20% of the samples are mislabeled by replacing their label uniformly at random). **(Right) DIVA for outlier rejection.** We use DIVA on a ResNet-34 network linearization and detect mislabelled samples (outliers) in a dataset present with 20% label noise. Selection is based on $\nabla_\alpha(L_{\text{val}}(\mathbf{w}_\alpha))_i > \epsilon$.

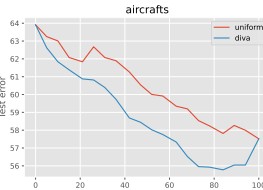 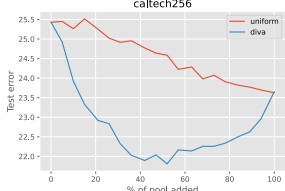 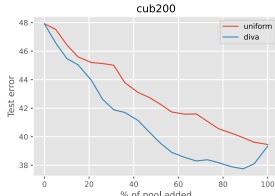

Figure 4: **DIVA Extend.** We show the test error achieved by the model as we extend a dataset with samples selected from a dataset pool using either DIVA Extend (red line) or uniform sampling (blue line). The pool set matches the same distribution as the training set. In all cases DIVA Extend outperforms uniform sampling and identifies subsets of the pool set with better performance than the whole pool. We also note that using only a subset selected by DIVA as opposed to using the whole pool, actually improves the test accuracy.

**Multi-modal learning.** Recent multi-modal models such as CLIP (Radford et al., 2021) can embed both text and images in the same vector spaces. This allows to boost the performance on few-shot image classification tasks by also adding to the training set textual descriptions of the classes, such as the label name. However, training on label names may also hurt the performance, for example if the label name is not known by the CLIP model. To test this, we create a few-shot task by selecting 20 images per class from the Caltech-256 and use DIVA Extend to select an increasing number of labels to add to the training set. In Figure 5 (right) of the Appendix, we show that DIVA can select the beneficial label embeddings to add in order to improve the few-shot test performance. However, when forced to add all labels, including detrimental ones, the test error increases.

**Data augmentation.** To further test the versatility of DIVA, we qualitatively evaluate DIVA Reweight on the task of tuning the probabilities with which we apply a given data augmentation procedure. Let $t_1, \ldots, t_K$ be a set of data augmentation transformations. Let $\mathcal{D}^{t_k}$ be the result of applying the data augmentation $t_k$ to $\mathcal{D}$. We can create an augmented dataset $\mathcal{D}^{\text{aug}} = \mathcal{D} \cup \mathcal{D}^{t_0} \cup \ldots \cup \mathcal{D}^{t_K}$, by merging all transformed datasets. We then apply DIVA Reweight on $\mathcal{D}^{\text{aug}}$ to optimize the weight $\alpha$ of the samples. Based on the updated importance weights we estimate the optimal probability with which to apply the transformation $t_k$ as $p_k = (\sum_{i \in \mathcal{D}^{t_k}} \alpha_i)/(\sum_i \alpha_i)$. In particular we select common data augmentation procedures, horizontal flip and vertical flip, and we tune their probability on the Street View House Number, Oxford Flowers and the Oxford Pets classification tasks. We observe that DIVA assigns different probabilities to each transformation depending on the task (Figure 5 in Appendix): on the number classification task DIVA penalizes both vertical and horizontal flips, which may confuse different classes (such 2 and 5, 6 and 9). On an animal classification task (Oxford Pets) DIVA does not penalize horizontal flips, but penalizes vertical flips since they are out of distributions. Finally, on Flowers classification DIVA gives equal probability to all transformations (most flower pictures are frontal so all rotations and flips are valid).

## 5 DISCUSSION

In this work we present a gradient-based method to optimize a dataset. In particular we focus on sample reweighting, extending datasets, and removing outliers from noisy datasets. We note that by developing the notion of a dataset derivative we are capable of improving dataset quality in multiple disparate problems in machine learning. The dataset derivative we present is given in closed-form and enables general reweighting operations on datasets based on desired differentiable validation losses. In cases where a set-aside validation loss is not available we show the use of the leave-one-out framework enables computing and optimizing a dataset "for free" and derive the first closed-form dataset derivative based on the LOO framework.

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
