# OpenReview forum: "DIVA: Dataset Derivative of a Learning Task"
_ICLR.cc/2022/Conference — ICLR 2022 Poster_

### Official Review · Reviewer_9tXK · 2021-10-26

**Correctness:** 3
**Technical Novelty And Significance:** 3
**Empirical Novelty And Significance:** 2
**Recommendation:** 6
**Confidence:** 3

**Main Review:**

Pros:
- The paper is well-written and easy to follow.
- Learning sample weights without a separate validation dataset is interesting and useful since it enables us to use the full dataset as training data. Although there are several methods to learn sample weights as described in section 2, they assume a separate validation dataset.
- The authors derive the closed-form expression of the derivative of the loss w.r.t. sample weights and it is attractive in terms of efficiency.

Cons.
- As described in the Introduction, the proposed method assumes pre-trained models to obtain the closed-form expression of the derivative, which might restrict its applicability in some cases. Is it possible for the proposed method to optimize model (neural network) parameters and sample weights simultaneously?
- Although the authors conducted experiments in multiple tasks, it is not sufficient. For example, comparison methods are a little naive. Comparison with other methods was conducted only in the dataset autocuration experiment.

Other comments and questions:
- Although the authors explicitly derive the Jacobian of $w_{\alpha}$ in eq. (8), is it sufficiently fast by using automatic differentiation of $w_{\alpha}$ (eq. (5))?
- How important is the initialization of sample weights $\alpha$?
- Although the proposed method learns sample weights without additional validation data, is it also possible to tune its hyperparameters such as optimization iteration numbers without the validation data?
- It is better to describe how many trials were performed in each experiment.
- It is unclear which loss was used ($L_{val}$ and $L_{LOO}$)  in the main paper.

Typo:
- second line of Introduction: how to select from from -> how to select from.
- $z_i$ -> $x_i$ in eq. (13)?
- in Figure 4, the meaning of blue and red lines would be opposite


**Summary Of The Paper:**

This paper proposes a method for dataset optimization that learns sample weights without a separate validation dataset.
By using squared loss and linearization around a pre-trained model, derivative w.r.t. the sample weights can be obtained as closed-form, which realizes efficient end-to-end learning. In addition, this paper uses leave-one-out cross-validation (LOOCV) loss instead of validation loss on a separate dataset, which enables to conduct evaluation without a separate validation dataset.
The experiments are conducted with multiple tasks (Dataset AutoCuration, Dataset extension, detrimental sample detection, multi-modal learning, and data augmentation).

**Summary Of The Review:**

Although the motivation of this work is interesting and the paper has some technical novelty, experiments can be improved.
Thus, I'm slightly leaning towards the reject side at this moment.

---After rebuttal---
Thanks for the response.
I think there is room for improvement in the experiment (comparison with existing methods for each task),
but considering the promise of the proposed method in a number of tasks and other reviews, I decide to increase my score to 6.

---

> ### Author Response · Authors · 2021-11-23
> **Response to Reviewer 9tXK**
>
> We thank the reviewer for their response and thoughtful questions. We address the specific questions below:
>
> ***Is it possible for the proposed method to optimize model (neural network) parameters and sample weights simultaneously?***
>
> Note that we are optimizing alpha and w at the same time: The optimization of DIVA runs bi-level optimization. In the inner loop, for a fixed alpha, we find the parameters that minimize  the weighted training loss (using the closed-form expression of the minimum), then in the outer loop the dataset weights (alpha) are optimized with respect to the validation loss. Hence, both the sample weights and the network parameters are optimized, but in alternating steps. The two steps are, however, merged into one in eq. 10 since it leads to a more efficient implementation. In the response to Reviewer 2K7t, we note that alternating small gradient steps in alpha and w may lead to suboptimal and degenerate solutions due to the non-convexity of the joint optimization.
>
> ***Although the authors explicitly derive the Jacobian of wα in eq. (8), is it sufficiently fast by using automatic differentiation of  wα  (eq. (5))?***
>
> The explicit derivative of w w.r.t to alpha that we write in closed-form in eq. (8) is indeed not computed explicitly when we compute the derivative of L_val w.r.t to alpha. Instead eq. (10) is used since the dataset derivative w.r.t to the loss is what is required for DIVA and using an efficient implementation of eq. (10) leads to very fast computation time. Since we have the derivative in closed-form we do not need to use automatic differentiation, which turns out to be less efficient.
>
> ***How important is the initialization of sample weights α?***
>
> We use the initialization of the weights alpha to reframe the problem: When looking at dataset reweighting, we always start with alpha_i = 1 for all samples and then optimize. For dataset extension, we initialize the core dataset samples to alpha_i = 1 and the pool dataset samples to alpha_i = 0, then by running dataset extension, we identify which samples with 0 weight should be added to the core training set. Mathematically, for different values of initial alpha, w_alpha will have different solutions leading to a different dataset derivative. It could be interesting to see however if using initialization with alpha_i  which is not equal to 1, can be beneficial in some cases.
>
> ***Although the proposed method learns sample weights without additional validation data, is it also possible to tune its hyperparameters such as optimization iteration numbers without the validation data?***
>
> We found that DIVA is robust to choices of hyper-parameters. In particular, for all the fine-grain datasets we present we use the same number of gradient update steps, and step-size. This makes the method ready to be used on different datasets without a separate validation set.
>
> ***It is unclear which loss was used (Lval and  LLOO) in the main paper.***
>
> We now added in the description of all tables and figures if DIVA LOO or DIVA val is being used. For Table 1, Figure 4, and Figure 5 DIVA LOO is being used.

---

### Official Review · Reviewer_3qBW · 2021-10-28

**Correctness:** 4
**Technical Novelty And Significance:** 3
**Empirical Novelty And Significance:** 4
**Recommendation:** 8
**Confidence:** 4

**Main Review:**

**List strong and weak points of the paper. Be as comprehensive as possible.**

**Pros:**

* The introduction is well written and motivates the necessity of DIVA and discusses its limitations.
* the related work sectiton does a great job at showing the similarities and differences compared to other approaches.
* Data quality is often a major problem and solved with overparametrized models. Tackling this issue with a closed-form solution is great.
* I think it's an important contribution to raise awareness for the limitations of only one predefined validation set. For computational reasons, deep learning is often not using the previously well established concept of k-fold cross-validation, and this paper offers an attractive alternative for more robust evaluation of neural networks.

**Cons:**

* There are a few question left open, see questions below.
* The empirical behavior off DIVA is only shown on image data. We do not know how it performs in other domains or types of datasets.
* Minor/Typos:
  * Introduction line 2: "from from"
  * Introduction last paragraph before summary: "LQF" should be defined
  * Figure 5 (right): "Test Error (\\%)" remove "\\"
  * Equation 13: ",." -> only "," should be there?

**Clearly state your recommendation (accept or reject) with one or two key reasons for this choice.**

I think this is a valuable and interesting contribution and I vote for accept. Going beyond a single fixed validation in deep learning set is important and often neglected and DIVA proposes an efficient approximation for a better estimate of the generalization error.

**Ask questions you would like answered by the authors to help you clarify your understanding of the paper and provide the additional evidence you need to be confident in your assessment.**


* it would be helpful to have maybe a piece of pseudocode describing how the application of DIVA works.
  * Is it necessary to train the linear layer?
  * Does this layer represent the classification task?
* The paper only shows results for classification tasks. Is it possible to apply DIVA for regression or other tasks in general? Would be helpful to add some clarification and at least mention other tasks.

* the authors mention that

  *For our experiments on dataset optimization we consider datasets that are smaller than the large scale datasets used for pre-training as we believe they reflect more realistic conditions for dataset optimization*

  it seems that computational efficiency could also be an important factor with large data sets, right?

* the idea of efficient leave-one-out errors is also well known in the kernel machine literature and it seems that some of the results are similar to the results for efficient leave-one-out crossvaliation for e.g. kernel ridge regression or kernel logistic regression (see e.g. [Cawley and Talbot, "Efficient approximate leave-one-out cross-validation
for kernel logistic regression", Machine Learning Journal, 2008](https://link.springer.com/content/pdf/10.1007/s10994-008-5055-9.pdf)). I don't think this similarity weakens the contribution of this submission, but it could be helpful to comment on the relationship between the dual variables in kernel machines and the importance weights in the DIVA approach

* sorry if I missed that, but would it be possible to include a direct comparison of the approximation and the true LOO error? The SGD based optimization seeems a bit unconventional, with the step size and very early stopping. Maybe some of the solutions from the kernel literature or matrix algebra, e.g. using cholesky decompositions, would be helpful for more efficient optimization of DIVA?

**Provide additional feedback with the aim to improve the paper. Make it clear that these points are here to help, and not necessarily part of your decision assessment.**

A detailed description how to apply DIVA would improve replicability.

Also it would be helpful to see the computational complexity of DIVA as a function of the data set size.

And I'm not sure I fully understand the argument with DIVA not needing a validation set. Leave-one-out cross-validation (LOO CV) is using validation data, it's just split differently. Ideally every model should report k-fold cv'd results, and the estimate of the generalization error will become better the larger k. The fact that often researchers use a predefined validation set is probably more due to the high computational complexity of doing CV properly. So I think I wouldn't emphasize that aspect too much as an advantage of DIVA, that it's applicable when there is no validation set - k-fold CV/jackknife/LOO would also help in that case.

**Summary Of The Paper:**

The paper introduces an approach to calculate the derivative of data samples' influence on the validation error. For this, the authors use a recent model linearization method and leverage closed-form leave-one-out loss calculation for linear models. Besides the theoretical derivation, they present consistent improvements in dataset extension, re-weighting, outlier rejection and automatic aggregation of multi-modal data.

**Summary Of The Review:**

This paper is a valuable and interesting contribution. It raises awareness for the importance of going beyond a single validation split for evaluation of neural networks and presents an interesting closed form solution for the leave-one-out (LOO) error that is, to the best of my knowledge, novel. The similarities of the proposed approach with efficient LOO cross-validation approaches for kernel machines suggests that there could be synergies that would allow to improve the optimization procedure of DIVA.

---

> ### Author Response · Authors · 2021-11-23
> **Response to Reviewer 3qBW (Part 1)**
>
> We thank the reviewer for their response and thoughtful questions. We also fixed all of the typos raised by the reviewer. We address the specific questions below:
>
> ***Is it necessary to train the linear layer?***
>
> No. Since we use an L2 loss, we use a closed-form expression for the optimal parameters of the linear layer of the task. We use this form implicitly in the update equation for alpha.
>
> ***Does this layer represent the classification task?***
>
> The last layer represents the classification task, which is used in the linearization of the manuscript experiments.
>
> ***The authors mention that "for our experiments on dataset optimization we consider datasets that are smaller than the large scale datasets used for pre-training as we believe they reflect more realistic conditions for dataset optimization it seems that computational efficiency could also be an important factor with large data sets, right?***
>
> Computational efficiency is indeed important for large datasets, but our derivation for DIVA is scalable to those settings as well. When the dataset is large, the DIVA derivative is computed efficiently and amounts to one forward pass of the network through the entire validation set, and a forward pass + storing the model predictions on the entire training set. The added computational costs of computing the DIVA derivative still enables optimization on large scale datasets. We do note that the linearization is faithful if the model weights are in the proximity to their pre-trained values. This however, does not deter from applying the algorithm in settings with large datasets and many training steps.
>
> ***the idea of efficient leave-one-out errors is also well known in the kernel machine literature and it seems that some of the results are similar to the results for efficient leave-one-out cross-validation for e.g. kernel ridge regression or kernel logistic regression (see e.g. Cawley and Talbot, "Efficient approximate leave-one-out cross-validation for kernel logistic regression", Machine Learning Journal, 2008). I don't think this similarity weakens the contribution of this submission, but it could be helpful to comment on the relationship between the dual variables in kernel machines and the importance weights in the DIVA approach***
>
> Thank you for bringing this up, we review the foundational works of Cawley in the related work section. Compared with the paper in the link, there are a couple of points we would like to highlight.
>
> * The dual variables in the kernel are also denoted by alpha and are matched to each data point, however, they turn out to be different. The dual variables alpha in (Cawely & Talbot) are the component weights of the featurized training samples used to create the model's predictions, . i.e. this is coming from the kernel trick, with alpha being the weight of each feature vector. In particular the iterative weighted least squares presented is just the method of optimization as opposed to a weighting scheme on the data samples.
> * In (Cawely & Talbot) the authors then use the LOO loss to run model selection, e.g. select the ridge parameter or other kernel hyper-parameters. This approach is a more traditional hyperparameter selection/optimization, whereas in our paper, we run optimization on tens of thousands of parameters, and use the gradient to inform data-centric decisions such as the selection of samples, reweighting, or dropping outliers. This is a completely different flavor but we do note that DIVA can be seen from the AutoML framework similarly to other model selection methods.

---

> > ### Author Response · Authors · 2021-11-23
> > **Response to Reviewer 3qBW (Part 2)**
> >
> > (Continued)
> >
> > ***sorry if I missed that, but would it be possible to include a direct comparison of the approximation and the true LOO error? The SGD based optimization seems a bit unconventional, with the step size and very early stopping. Maybe some of the solutions from the kernel literature or matrix algebra, e.g. using cholesky decompositions, would be helpful for more efficient optimization of DIVA?***
> >
> > * DIVA uses two separate optimizations: the outer one is an optimization of the sample weights alpha, which as the reviewer says uses gradient descent with large learning rate and early stopping. The inner optimization is instead a more standard optimization of the parameters of the classifier given alpha. For this, we directly use the exact closed-form expression of the optimum which indeed we compute using a cholesky decomposition so no steps of gradient descent are needed.
> >
> > * For the outer optimization of alpha, we need to optimize using a generic gradient descent scheme since the loss function is non-convex and may not have a closed form solution. The early stopping here is necessary since we optimize a very large number of parameters with respect to the validation set, which leads to overfitting after a large number of gradient updates (more formally, as we mention in the paper, the validation loss is not an unbiased estimator of the test loss after the first step). We also note that if computational time is a concern, applying even just a single gradient update step for alpha is enough to give improved results, for Table 1, the average improvement for a single step of DIVA is +1.21.

---

### Official Review · Reviewer_7aEq · 2021-11-08

**Correctness:** 3
**Technical Novelty And Significance:** 4
**Empirical Novelty And Significance:** 4
**Recommendation:** 8
**Confidence:** 3

**Main Review:**

In general, I think this is a good paper.

I think it addresses a longstanding and open question -- how do we know we are feeding our models the right data during training to best set them up for success?  This paper adds a new tool in the arsenal for figuring out which data is best for a given learning task.  I believe the focus of this paper would be of interest to a broad audience.

The discussion section ends very abruptly.

I would like to see this method extended to apply in cases where the model class in question is not a DNN.  The Dataset Extension, Curation, and Reweighting capabilities would also be of great use for models trained on tabular data.

One thing that feels missing in the paper is a good real-world example of how this method could make positive impact.  i.e. an immediate thought that comes to mind could be a scenario where multiple domain experts label data, but inter-rater reliability may be low - using DIVA to downweight samples which ultimately confuse the model seems useful in a case like this, which often arises in clinical contexts.

A relationship that I would like to see the authors explore in the discussion section is overfitting of the model.  DIVA reduces the weights of hard/confusing images (figure 3.4).  An alternate hypothesis for why the updated model performs better on test data is that the easy/canonical images can be adequately classified by a simpler model.  Side-by-side, the simple model trained on easy data may generalize better than a complex model trained on hard data.  Ruling this out would make DIVA appear stronger, as this would mean it is impossible to improve test set accuracy by simply feeding a model the most canonical images during training.

**Summary Of The Paper:**

The authors define a data set derivative, which can be used to update weights of individual samples in training data such that a trained model may increase its performance on test data.  In addition to re-weighting samples within a training data set, the authors also describe how the same techniques can be used to select new data points from a pool of potential candidates and used to estimate which samples are likely to be mislabeled in data.

**Summary Of The Review:**

I think the paper is in good enough shape to warrant publication with only minor changes.  I believe the focus of this paper would be of sheer interest to a broad audience.

---

> ### Author Response · Authors · 2021-11-23
> **Response to Reviewer 7aEq**
>
> We thank the reviewer for their thoughtful response and valuable suggestions. We address the specific questions below:
>
> ***an immediate thought that comes to mind could be a scenario where multiple domain experts label data, but inter-rater reliability may be low - using DIVA to downweight samples which ultimately confuse the model seems useful in a case like this, which often arises in clinical contexts.***
>
> We agree that this is an interesting example where DIVA could be very helpful. In particular, having multiple sets of labels for each image, a model can be trained to predict a convex combination of them. In this case, DIVA could be used to assign the weight assigned to each annotator in order to improve the final generalization of the model:
> \arg\min_\alpha_1 \alpha_2  L_train(w, \alpha_1, \alpha_2) = \sum_i \ell_w(x_i, \alpha_1 * y_i^(1) + \alpha_2 * y_i^(2))
> This may, for example, downweight incorrect annotations, or upweight annotators that implement correct and robust labeling strategies. While we do not currently have access to datasets labeled with labels from multiple sources, we note that in Figure 3, DIVA can discern mislabelled examples and that it can downweight confusing examples supporting this use case.
>
> ***A relationship that I would like to see the authors explore in the discussion section is overfitting of the model. DIVA reduces the weights of hard/confusing images (figure 3.4). An alternate hypothesis for why the updated model performs better on test data is that the easy/canonical images can be adequately classified by a simpler model. Side-by-side, the simple model trained on easy data may generalize better than a complex model trained on hard data. Ruling this out would make DIVA appear stronger, as this would mean it is impossible to improve test set accuracy by simply feeding a model the most canonical images during training.***
>
> Which sample weights would lead to the best generalization on a set aside test set is indeed the question at the heart of our method. This question deviates from the traditional empirical risk minimization as we acknowledge that:
>
> 1. The training set collection process may have led to unbalanced classes or different annotation rates for each class.
>
> 2. The model may generalize better when trained on a distribution that differs from the actual test set distribution.
>
> DIVA illustrates indeed that training with certain subsets of the dataset can lead to better test performance than training with all of the dataset, even for fairly clean datasets, (see Figure 4). Along the same vein, non-uniform reweighting of the training dataset also leads to improved test performance (see Table 1). While it is not clear to us what makes a training set better for a learning task and generalization, by optimizing with respect to the LOO and benchmarking on a separate unweighted test set, we see that the approaches presented for DIVA are not overfitting as the test set is completely untouched during the optimization. More generally, DIVA is a step in the direction of more interactive and flexible notions of learning from data in the DNN optimization framework as opposed to standard empirical risk minimization.

---

### Official Review · Reviewer_2K7t · 2021-11-09

**Correctness:** 4
**Technical Novelty And Significance:** 2
**Empirical Novelty And Significance:** 2
**Recommendation:** 5
**Confidence:** 5

**Main Review:**

Positives
+ The problem formulation and closed form solution to the dataset derivative are potentially interesting and generally useful for deep learning
+ The work achieves good results with respect to other previous works, and experiments on dataset expansion are very interesting

Negatives

My concerns with the work are to do with the methodology and experimental design choices are around comparison with sufficient baselines.

Methodology: The proposed experiments in the paper all use the deep network as a feature extractor and the last linear layer of the network to make decisions on classification. Thus, we are in a pretraining regime (where there is no dataset derivative being used) and a downstream task where we extract the pretrained features and do linear classification. Given this, the experimental / methods section (sec. 3) is very terse and complicated since it talks about linearizing the model etc. where the model actually that is used is infact already linear. I understand it might have been written that way with generality in mind, but the experiments should then also reflect that and demonstrate that the model / approach does indeed work for the deep learning case (say training from raw pixels) or something more general like that.

Comparison to baselines: A number of important comparisons to baselines / discussion around those approaches seem to be missing in this current work.

1) The paper mentions that it is difficult to compare to Koh and Liang (2017) which directly addresses the problem of outlier detection and also the problem of dataset reweighting for better genrealization by estimating the influence of upweighting a data sample by epsilon. The paper mentions that the work was not compared to because of computational reasons, but I am struggling to see how that work is much more expensive than the current work or the work of Ren et.al. (2018). For example, given the task of dataset reweighting, it appears to me that the work of Koh and Liang requires the following to be computed: 1) the hessian of the data at the optimum solution on the training set (unweighted), 2) the gradient of the loss with respect to the parameters at the training datapoints, and 3) the gradient of the loss with respect to the parameters at the test datapoints. 1) and3) are required for the proposed approach as well, and 2) is not that expensive to compute for a Mean Square Error Regression. Secondly, unlike the claim in the paper Koh and Liang can be used to do dataset reweighting by computing the influence of each training datapoint on the validation set, and keeping around some top-K training samples and checking how well one does based on them.  (*)

2) Another important paper in this literature which is missing (and should really serve as a baseline) is that of [A]. I think this work should definitely be compared to the current framework. (*)

3) As far as comparison to Ren. et.al. (2018) is concerned, the key difference is that the current work proposes to linearize the classifier function (whereas Ren et.al. (2018)) use a linear approximation to the actual loss function, since they do gradient descent whose variational form is essentially \min_{\theta’) L(\theta’) + || \theta - \theta’||_2^2 which is then approximated as \min_{\theta’} L(\theta) + d/d\theta (L(\theta)) (\theta - \theta’) + ||\theta - theta’||_2^2, solving which gives us the update equation for SGD. The second difference is that while Ren et.al. (2018) solve the update for the dataset in a single step and the inner loop problem in a single step (not fully), the current paper considers a linear approximation to the original problem and solves the two problems exactly. It would be nice to understand given this where exactly the gains of the work over that of Ren. et.al. (2018) come from. This can be achieved in two ways:

1) Solving the current problem sub-optimally by only doing one step of gradient descent and computing inexact solutions in the linear case, even though we know exact solutions (derived in this work exist). This would help to contextualize how much benefit we have from the current approach.
2) Solving a problem that is non-linear (as opposed to fixed feature extraction and linear classification) such that the linear approximation using taylor series actually comes into play in the sense that there are higher-order terms which are being ignored. In this setting it would be important to compare the current approach to Ren. et.al. This would tell us whether one benefits in this problem domain by solving the correct problem approximately (which Ren. et.al. do) or solving an approximate problem correctly (which this paper does) and what the tradeoff is between these modes. (*)

[A]: Lorraine, Jonathan, Paul Vicol, and David Duvenaud. 2019. “Optimizing Millions of Hyperparameters by Implicit Differentiation.” arXiv [cs.LG]. arXiv. http://arxiv.org/abs/1911.02590.

**Summary Of The Paper:**

The paper proposes a novel method for computing the derivative of the weights assigned to datapoints in a dataset with respect to the generalization error of a learning task. The core idea being that one can take a pretrained model, and compute its derivative with respect to a fine-tuning task/ dataset and understand which datapoints are important and which datapoints are not important. This can then be used for tasks such as 1) outlier detection, 2) dataset expansion, and 3) dataset reweighting. Comparisons of the proposed work to other previous work reveals that the current work achieves better results on dataset reweighting.


**Summary Of The Review:**

This is a promising method to the well studied problem of identifying the weighting of the datapoints in a learning task that aid generalization on a given task. The current method while promising is missing some vital comparisons which limit the ability of a reader to assess the true impact of the work, along with experiments which make a lot of the methods sections and details in it seem like overly complicated detail. Given this I do not think the current paper is ready for publication. Important points for the rebuttal are marked with a (*).

---

> ### Author Response · Authors · 2021-11-23
> **Response to Reviewer 2K7t (Part 1)**
>
> We thank the reviewer for the detailed response and interesting suggestions. Below we address the specific comments:
>
> ***The paper mentions that it is difficult to compare to Koh and Liang (2017) which directly addresses the problem of outlier detection and also the problem of dataset reweighting for better generalization by estimating the influence of upweighting a data sample by epsilon. The paper mentions that the work was not compared to because of computational reasons, but I am struggling to see how that work is much more expensive than the current work or the work of Ren et.al. (2018).***
>
> The method of Koh computes the influence of *individual* samples using an approximation of the Hessian based on conjugate-gradients. Making the method scale to the point that the influence of all samples can efficiently be computed in batch mode, as needed for dataset optimization, requires precisely the closed-form expression for the derivative that we present in the paper (eq. 10). Ren et al. also notice the same problem with the computational costs, which prompted them to develop their approximated algorithm, where they bypass the computation of the CG approximation of the Hessian and instead use a cheap single-step estimate to find the sample influence. Lastly it is not clear to us how one would extend the work of Koh to use the leave-one-out-validation (LOOV)  loss that we present in Section 3, Table 1.
> We also note that, while to derive their influence function the work uses the epsilon upweighting argument, the work of Koh et al. does not run dataset reweighting optimization, and the work is concerned with 1) label flipping 2) constructing poisoning attacks 3) removing outliers. So there is no dataset reweighting, or dataset extension in the work of Koh.
>
> ***[A]: Lorraine, Jonathan, Paul Vicol, and David Duvenaud. 2019. “Optimizing Millions of Hyperparameters by Implicit Differentiation.” Another important paper in this literature which is missing (and should really serve as a baseline) is that of [A]. I think this work should definitely be compared to the current framework.***
>
> We thank the reviewer for suggesting this relevant work, which we will add and discuss in the related work section.  The method computes an approximate gradient of a generic model/loss with respect to a generic set of hyper-parameters and shows application to important areas as neural-network based augmentation, dropout, and weight decay tuning. However,  in our setting the hyper-parameters are the sample weights in which case we can compute directly the exact gradient in a more efficient and straightforward implementation (eq. 10). Moreover their method would not work when using the leave-one-out-validation loss instead of a separate validation set, which is one of the main contributions of DIVA.
>
> ***As far as comparison to Ren. et.al. (2018) is concerned, the key difference is that the current work proposes to linearize the classifier function (whereas Ren et.al. (2018)) use a linear approximation to the actual loss function, since they do gradient descent whose variational form is essentially \min_{\theta’) L(\theta’) + || \theta - \theta’||2^2 which is then approximated as \min{\theta’} L(\theta) + d/d\theta (L(\theta)) (\theta - \theta’) + ||\theta - theta’||_2^2, solving which gives us the update equation for SGD. The second difference is that while Ren et.al. (2018) solve the update for the dataset in a single step and the inner loop problem in a single step (not fully), the current paper considers a linear approximation to the original problem and solves the two problems exactly. It would be nice to understand given this where exactly the gains of the work over that of Ren. et.al. (2018) come from.***
>
> We believe that the main reasons for the method of Ren et al. is underperforming are:
>
> 1. They do not actually provide a way to correctly solve the linearized problem (eq. 6 in their paper). Rather they suggest a simpler estimate in eq. 8 that sets the weight of the examples equal to normalized and thresholded processing of the gradient of epsilon (see line 11, of Algorithm 1 in Ren et al. https://arxiv.org/pdf/1803.09050.pdf#page=4 ). On the other hand, not only does DIVA use the linearization of the model, it also uses the exact gradients and we take multiple gradient steps to optimize alpha, to get a more accurate estimation of the optimal weights.
>
> 2. The optimization is done on small mini batches. The optimal weighting of the small batch that they consider may not reflect the optimal weight when the sample is considered together with the whole dataset. This is not a problem in their setting, since the examples that they downweight are clear outliers (e.g. wrong labels of a noisy dataset) and easy to exclude. On the other hand, in most of our tasks, there are no outright wrong samples and reweighting correct samples with an inaccurate estimate may do more damage than good.

---

> > ### Author Response · Authors · 2021-11-23
> > **Response to Reviewer 2K7t (Part 2)**
> >
> > (Continued)
> > ***Solving the current problem sub-optimally by only doing one step of gradient descent and computing inexact solutions in the linear case, even though we know exact solutions (derived in this work exist). This would help to contextualize how much benefit we have from the current approach.***
> >
> > In DIVA we cannot take a single gradient step in the weights, as the method implicitly uses the closed-form of the optimal weights. We can, however, take a single gradient step to update alpha, which would make it closer to the reweighting methods suggested in eq. (6). We tried comparing our multi-step approach with a single-step approach, and the gain drops from +2.29 to +1.21. The worse performance suggests that multiple-steps are important. However, since the method still improves the test performance, it  suggests that the other heuristics of Ren at al., together with the use of mini-batches (see above), may be responsible for their method under-performing on our use-cases.
> >
> > Finally, we also would like to point out that, in addition to the several technical advantages both theoretical and computational of our method compared to [A], Koh et al. and Ren et al., our method is the only method that allows the use of the leave-one-out-validation loss to optimize. This last point is particularly important since the datasets that would benefit most from being expanded or augmented are small datasets for which cannot spare enough data for a separate validation set that is large enough to give a good estimate.

---

### Decision · Program_Chairs · 2022-01-20

**Decision:**

Accept (Poster)

**Comment:**

This paper has been independently reviewed by four expert reviewers. Two of them recommended straight acceptance, one of them assesses this work as marginally acceptable after increasing their score as a result of the author's rebuttal, and the last reviewer considers this paper marginally below the acceptance threshold. While the reviewers agree on the importance of the  targeted problem and relative novelty of the presented work, the main points of criticism involve empirical evaluations - its methodology, experimental design, missing relevant and important comparisons. Since the authors have addressed most of those concerns in their rebuttal, I am leaning towards recommending acceptance of this work for ICLR.